# Reversible polymer-gel transition for ultra-stretchable chip-integrated circuits through self-soldering and self-coating and self-healing

Pedro Alhais Lopes[1,2], Bruno C. Santos [1,2], Anibal T. de Almeida[1] & Mahmoud Tavakoli [1✉]

Integration of solid-state microchips into soft-matter, and stretchable printed electronics has been the biggest challenge against their scalable fabrication. We introduce, Pol-Gel, a simple technique for self-soldering, self-encapsulation, and self-healing, that allows low cost, scalable, and rapid fabrication of hybrid microchip-integrated ultra-stretchable circuits. After digitally printing the circuit, and placing the microchips, we trigger a Polymer-Gel transition in physically cross-linked block copolymers substrate, and silver liquid metal composite ink, by exposing the circuits to the solvent vapor. Once in the gel state, microchips penetrate to the ink and the substrate (Self-Soldering), and the ink penetrates to the substrate (Self-encapsulation). Maximum strain tolerance of ~1200% for printed stretchable traces, and >500% for chip-integrated soft circuits is achieved, which is 5x higher than the previous works. We demonstrate condensed soft-matter patches and e-textiles with integrated sensors, processors, and wireless communication, and repairing of a fully cut circuits through Pol-Gel.

[1] Soft and Printed Microelectronics Lab, Institute of Systems and Robotics, Department of Electrical Engineering, University of Coimbra, Coimbra, Portugal. [2]These authors contributed equally: Pedro Alhais Lopes, Bruno Santos. ✉email: mahmoud@isr.uc.pt

Stretchable electronics have a wide range of applications in flexible displays[1], sensors[2–6], health monitoring devices[7–12], structural electronics[13,14], wearable motion sensing[15], e-textiles[16–19], and implantables[20]. Materials and fabrication methods have been the focus of many research efforts during the past decade. Such efforts mostly focused on stretchable electrodes, interconnects, and sensors. However, the ultimate functionality of such systems remains dependent on solid-state technology (SST), from simple light-emitting diodes (LEDs) to packaged integrated circuits (ICs) for data acquisition, processing, and communication. Integration of SST into these circuits induces a drastic mechanical mismatch, resulting in premature failure and/or loss of the circuit functionality[15]. For instance, ultra-stretchable interconnects with elongation at break of up to 1000% were demonstrated[21,22], but the very few works that demonstrated microchip-integrated circuits could not withstand >100% strain[15,22]. In addition, independent of the maximum attainable strain, the main challenge is the complexity of the existing methods for integration of surface mount devices (SMDs) into stretchable circuits, which are labor-intensive, costly, and involve many fabrication steps. Addressing this challenge is the key to the scalable fabrication of stretchable circuits.

Seminal efforts for the fabrication of stretchable interconnects have focused on deterministic circuit architectures with circuit traces that have a wavy[23], horseshoe, or serpentine geometry. But to reach a chip-integrated circuit, this process requires over ten fabrication steps including cleanroom lithography[24]. As an alternative, conductive and printable composites were also demonstrated, using different blends of elastic polymers and conductive micro/nanoparticles, wires, or tubes[25,26]. Also, liquid metal (LM)— including eutectic gallium–indium (EGaIn; 75.5 wt% Ga and 24.5 wt% In), and gallium–indium–tin (Galinstan; 68 wt% Ga, 22 wt% In, and 10 wt% Sn) have been popular due to their fluidic compliance[27], high electrical conductivity ($\sim 3.5 \times 10^6$ S/m)[28], and their desirable behavior under strain, i.e., smaller gauge factor[29] and lower electromechanical hysteresis than conductive composites that typically suffer from the "Mullin's effect"[30]. Methods reported for patterning of LM-based circuits include injection molding[31], stencil lithography[32], selective wetting[33], reductive patterning[34], rollerball pen filled with LM[35], 3D printing[36,37], laser patterning[15,38,39], micro-contact printing[40], inkjet printing[41,42], selective LM plating[43], LM coating over printed Ag[44,45] and Au[46], high-resolution poly(vinyl alcohol)-mediated printing[47], pen[48,49], LM nanoparticle printing and selective wetting of over poly(methyl methacrylate) glue[50], direct digital printing of biphasic Ag–In–Ga composites[22], and transfer printing of biphasic EGaIn[51,52]. Reviews of some of these methods for digital printing of stretchable electronics can be found in recent literature[30,53]. In one work, component pick and place were well combined with printing conductive composites for hybrid 3D printing[37].

Despite these advances in patterning stretchable electrodes and interconnects, integration of SST chips into these circuits remained the main challenge in this field. Addressing this challenge is the key to scaling the production and commercialization of stretchable circuits.

Traditional soldering techniques cannot be used due to their incompatibility with the LM and also due to the heat sensitivity of many elastic substrates. Some works demonstrated methods to address the integration of SST into LM-based stretchable circuits with innovative techniques. This includes interfacing connections using HCl vapor-treated EGaIn "solder"[54] and a direct connection to a simple complementary metal–oxide–semiconductor die[55]. Indirect integration uses flexible printed circuit boards (FPCBs)[56] or z-axis conductive films[45,57,58]. In one example, a maximum strain of ~100% was achieved[58] using an engineered anisotropic conductor for interfacing the chips to the LM circuit,

but a protective sealing layer was necessary to hold the microchip in place. In the absence of this layer, the maximum strain was reduced from 100 to 30%.

Electrically conductive adhesives (ECAs) are popular for non-stretchable printed circuits, but ECAs demands high-precision selective deposition system[59], compatibility between the adhesive and the conductive ink, rapid SMD placement prior to the adhesive drying, precise motion control in component placement system, for avoiding the spread of the conductive adhesive, and a thermal sintering step.

The healing ability of some polymer networks is an interesting property that enables repairing damaged polymers. To do so, several methods had been explored, including solvent mediated[60] exposure or solvent vapor exposure[61–63]. Using these techniques, cracks can be healed and electrical and mechanical properties can be recovered by promoting the entanglement of polymer chains with heat above the glass transition temperature or promoting the diffusion of the material using a solvent.

Here, we present materials and methods that reduce the fabrication complexity of SST-integrated circuits. The process includes three simple steps, printing, component placing, and the polymer–gel (Pol–Gel). The reversible transition between the polymer state and the gel state (Pol–Gel) is performed by exposing the polymer to the vapor of its solvent. Styrene–isoprene block copolymers (SIS) are used as the substrate and as the elastomeric matrix of the conductive ink. SIS is used in this work due to its transparency, excellent elastic properties in the solid phase, and its strong adhesion properties in the liquid phase, but the process may be extended to other polymers and other types of stimulus for triggering the Pol–Gel transition. During the gel state, the microchip is surrounded by the adhesive polymer from five sides and the conductive ink adheres to the pads of the package. This "self-soldering" procedure allows seamless integration of microchips into soft circuits, resulting in a record-breaking maximum strain tolerance of >500%; it enhances the conductance of the printed interconnects more than 2 times through improved percolation of the micro fillers and it heals the microcracks of the substrate, thus improving the strain tolerance of the printed interconnects to ~1200% strain. Besides, during the exposure, an interesting self-coating over the printed traces occurs, which saves another fabrication step, i.e., application of the sealing layer.

It is important to highlight the role of block copolymers (BCPs) in the success of this method. Styrenic BCPs are generally composed of rubber chains that are physically cross-linked by the rigid polystyrene domains[64]. This provides a combination of elastic properties in the polymer state, and the possibility of the reversible Pol–Gel transition, through exposure to the polymer solvent.

In summary, this material architecture, and the proposed vapor exposure technique, allows autonomous soldering of the chips, and sealing of the printed circuits, in a unique production step. Supplementary Videos 1 and 2 show the side view and top view of the inside chamber footage of the Pol–Gel process for self-soldering and self-coating process.

The vapor exposure can also be used to embed the printed circuit onto other surfaces such as a textile. Finally, we show that after making a full cut on the printed circuit and the substrate, the vapor exposure is able to heal the circuit so effectively that it not only restores the electrical functionality but also can withstand the strain.

## Results

**Circuit fabrication.** The circuit starts by application of an ~100 μm SIS solution using a thin-film applicator. The desired circuit is printed using a desktop extrusion printer (Voltera V-one)

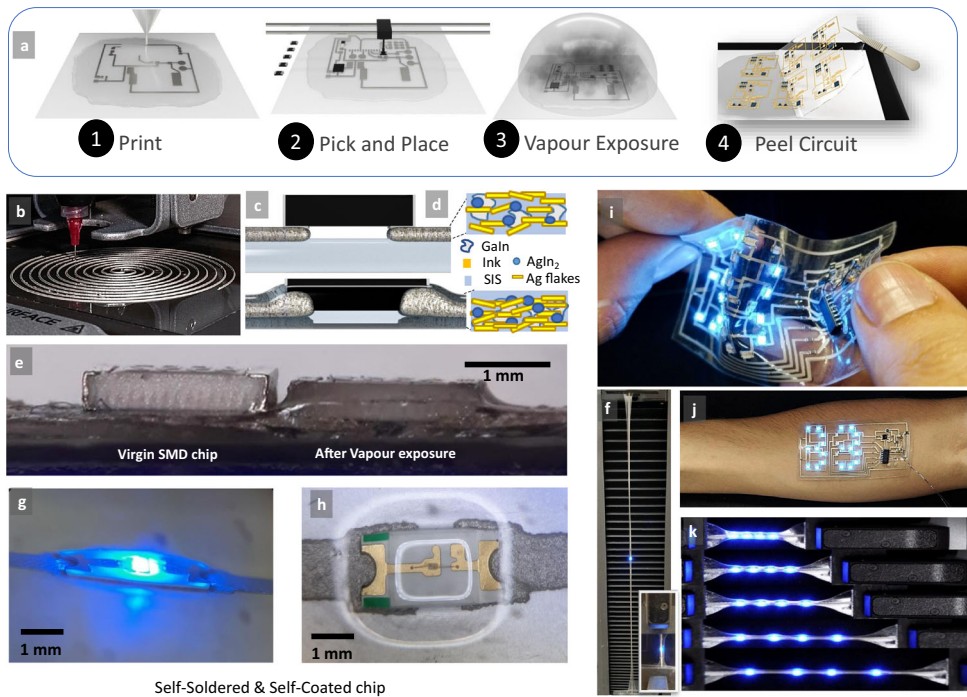

**Fig. 1 Fabrication process for hybrid stretchable circuits. a** Process for chip-integrated circuit fabrication. Printing using an extrusion printer and the Ag–In–Ga–SIS ink, pick and place of microchips, Pol–Gel transition through solvent vapor exposure, and peeling of the circuit. **b** Example of a digitally printed circuit with 14 layers of ink. **c** Schematics of the chip interfacing through Pol–Gel. **d** Schematics of the ink microstructure before and after vapor exposure. **e** Side view of a small resistor chip before and after vapor exposure. **f** A sample with an integrated LED, after being cut, and healed through Pol–Gel, <900% strain. **g** Side view of a self-encapsulated circuit through Pol–Gel. **h** Top view of a self-encapsulated circuit through Pol–Gel. **i** Soft-matter circuit with integrated sensors, microprocessor, and LED display for temperature measurement. **j** Same circuit was applied to the skin. **k** Example of a circuit with multiple LEDs under strain. The authors affirm that the human research participant (first author P.L.) provided informed consent for publication of the images in (**i**, **j**).

(Fig. 1 and Supplementary Fig. 1) using an Ag–EGaIN–SIS ink (see "Methods") that guarantees excellent electrical conductivity, a maximum strain tolerance >1200%, and stable electromechanical behavior over repeated strain cycles. SMD chips are then placed on the circuit using a pick and place machine. Afterward, the circuit is placed in a sealed chamber with toluene vapor for ~1 h and is finally peeled from the surface (Fig. 1a). Toluene vapor causes the Pol–Gel transition both on the SIS-containing printed ink and the underlying substrate. Both ink and the substrate turn into a soft and highly adhesive gel, owing to the adhesive properties of the SIS. The SST component penetrates slightly into the softened ink and the conductive pads of the component adhere to the gel-like adhesive ink. At the same time, the component and the ink penetrate together into the softened SIS substrate (Fig. 1c). As the ink descends into the SIS gel, a thin layer of the polymer covers the ink, which protects the ink over time (Fig. 1c, microscopy images of Supplementary Fig. 9, and Videos 4 and 6). In addition, in the gel state, SIS slightly ascends the walls of the microchip component and surrounds all four sides of the component perimeter, which seems to be related to the capillary effect. Figure 1e compares the side view optical image of an SMD resistor before and after toluene exposure. It is visible from this figure (also see Supplementary Videos 1–4) that the component descended into the substrate and also some SIS rises on the walls of the package. All these procedures are performed at room temperature and solely by exposing the circuit to the solvent vapor.

Overall, this self-soldering technique results in a seamless integration of the microchip into the ink and the substrate, without the need for selective deposition of conductive/insulator adhesives, and additional sintering process. The resulting circuit can withstand >500% strain prior to the electrical failure. Note that in contrast to previous works, no sealing layer is applied over the circuit to fix the components, and all of the procedure is performed at room temperature.

Vapor exposure also enhances the conductivity of the ink, owing to the improved percolation of the Ag–EGaIn (or more precisely Ag–In–Ga) microstructure. Figure 1f shows a chip-integrated sample <900% strain after the sample was cut and healed through the Pol–Gel method, as will be further explained. The inset in Fig. 1f shows the sample prior to the strain. Figure 1i, j shows a circuit with several SST microchips, including a temperature sensor, a microcontroller, and 20 LEDs, which display the skin temperature in real time. Figure 1k shows an example of a circuit with integrated LEDs under strain.

**Electromechanical characterization**. The coupling of the SST-integrated samples was analyzed by studying the maximum strain before electrical failure and also repeated strain cycles. Samples were produced in a dogbone shape (Supplementary Fig. 2) and interfaced with SMD resistor chips using the vapor exposure method. None of the samples benefited from a sealing layer. First, 11 samples with an integrated resistor package (0805) were stretched until electrical failure. We chose this package, compared with the previous works[57], for the characterization of chip-integrated stretchable circuits. Figure 2a (inset) shows that nine samples could withstand between 500 and 700% strain, one sample overpassed 700% strain, and another sample reached 900% strain. Although it was possible to reach 900% strain, factors related to imperfections in the manufacturing process, resulting in the premature breakdown <600% strain. When a high amount of strain is applied, any imperfection in the substrate, the printed trace, or on the alignment between the chip and the

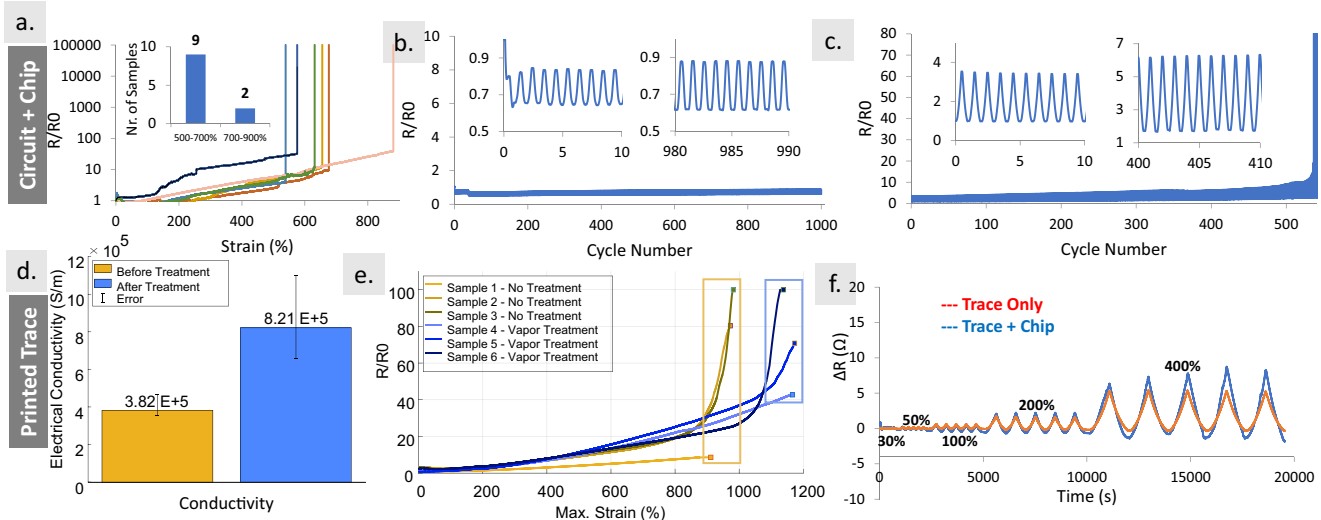

**Fig. 2 Electromechanical characterization. a** Max. strain tolerance of multiple samples with an integrated resistor package. **b** One hundred percent strain cycle of the chip-integrated sample. **c** Four hundred percent strain cycle of the chip-integrated sample. **d** Conductivity of printed traces before and after the Pol–Gel. **e** Maximum strain tolerance of the printed traces before and after the Pol–Gel. **f** Electrical resistance comparison of a printed trace without and with integrated chip, under strains of 30, 50, 100, 200, and 400%.

printed trace causes the creation of microcracks that are propagated until the mechanical failure. During the vapor treatment, if the surface over which the samples are placed is not perfectly flat and level, the SIS gel moves slightly to one side (Supplementary Video 4), thus resulting in some nonuniformity in the circuit as well. Nevertheless, a strain tolerance of >r 500% was easily attainable in all samples.

Unlike the previous works in which the package is only bonded from the bottom surface, this method restricts the package from five faces, which contributes to a better mechanical locking, thus a more uniform distribution of the stress at the rigid–soft interface, resulting in a considerably higher strain tolerance compared to previous works. In all samples, the mechanical failure happened prior to the electrical failure. We observed that the failure usually starts on the interface between the component and the substrate (Supplementary Fig. 3).

We then studied the electromechanical coupling and stability of the samples over repeated cycles. The sample could withstand 1000 cycles of 40% (Supplementary Fig. 4) and 100% strain (Fig. 2b) successfully. Note that, in $R/R_0$ from Fig. 2b, the resistor value is subtracted from the overall measured resistance value. In both cases, $R_0$ (measured resistance at 0% strain) and $R_x$ (measured resistance at $x$% strain, $x = 40$ or 100), as well as $\Delta R$ (in each 0–$x$% cycle), remained almost constant for the whole 1000 cycles. This is important for the proper functionality of digital circuits. On a 400% cycle test, the sample could withstand 540 cycles (Fig. 2c).

As discussed earlier, vapor exposure also improves the electromechanical properties of the ink and the substrate. We characterized printed samples without the microchip, before and after vapor treatment, in order to examine these improvements. Figure 2d shows the average electrical conductivity of the samples after toluene vapor exposure was increased from $3.8 \times 10^5$ to $8.21 \times 10^5$ S/m, an increase of more than 2 times (see Supplementary Fig. 5 for details). The vapor exposure softens the SIS matrix in the ink, which provides an opportunity for the Ag flakes and other microparticles to re-assemble themselves in a more compact form, resulting in a better 3D percolating network and higher conductivity. These results are supported by the scanning electron microscope (SEM) and will be discussed later.

Surprisingly, vapor exposure also improves the maximum strain tolerance of the printed traces from ~950 to ~1200% (Fig. 2e). The mean gauge factor of the interconnects is 0.96, calculated at 200% strain. This is because the vapor exposure heals the microcracks of the substrate. These microcracks are starting points for the propagation of the larger cracks, and their elimination contributes to the improvement of the maximum strain tolerance. Note that these are only printed traces without an IC. In order to investigate the role of the strain on the interface between the resistor and the trace, we compared a printed trace, with a chip-integrated sample for repeated cycles of 30–50–100–200–400% strain (Fig. 2f). Only when the strain reaches 400% there is a noticeable difference between the resistances of the two samples. This indicates that at this strain some instabilities develop on the interface between the microchip and the printed circuit, which is consistent with the result of Fig. 2c.

**Microstructure analysis**. SEM and energy-dispersive X-ray spectroscopy (EDS) are used to examine the substrate and the material distribution of the printed ink, as well as the microchip component before and after vapor exposure. First, by analyzing images from the SIS substrate (Fig. 3a, b), we observed that microcracks and other imperfections that seem to be undissolved SIS on the surface of the substrate fully disappeared after vapor exposure. This should be the main reason for the improved maximum strain tolerance of the samples. Additional images with different magnification scales are provided in Supplementary Fig. 6.

Figure 3c, d shows secondary electron (SE) analysis of a sample before and after treatment, and Fig. 3e, f shows the backscattered electron (BSE) analysis of the same. The time of vapor exposure was reduced, so that the ink is not coated by the SIS, to allow microscopy analysis. The Ag flakes are visible on all samples. We also observe microspheres (<5 μm) that are $AgIn_2$ based on EDS analysis (see Supplementary Fig. 7). These microspheres are formed during the ink mixing and are present on both non-treated and treated samples. However, it seems that the treated samples appear more on the top surface, as can be seen in the SE images (Fig. 3c, d). In contrast, EGaIn droplets are only visible on the BSE images, which suggests that they are mostly buried below Ag micro flakes and $AgIn_2$ microparticles. It is worth mentioning that BSE analysis is able to scan deeper layers of the sample when

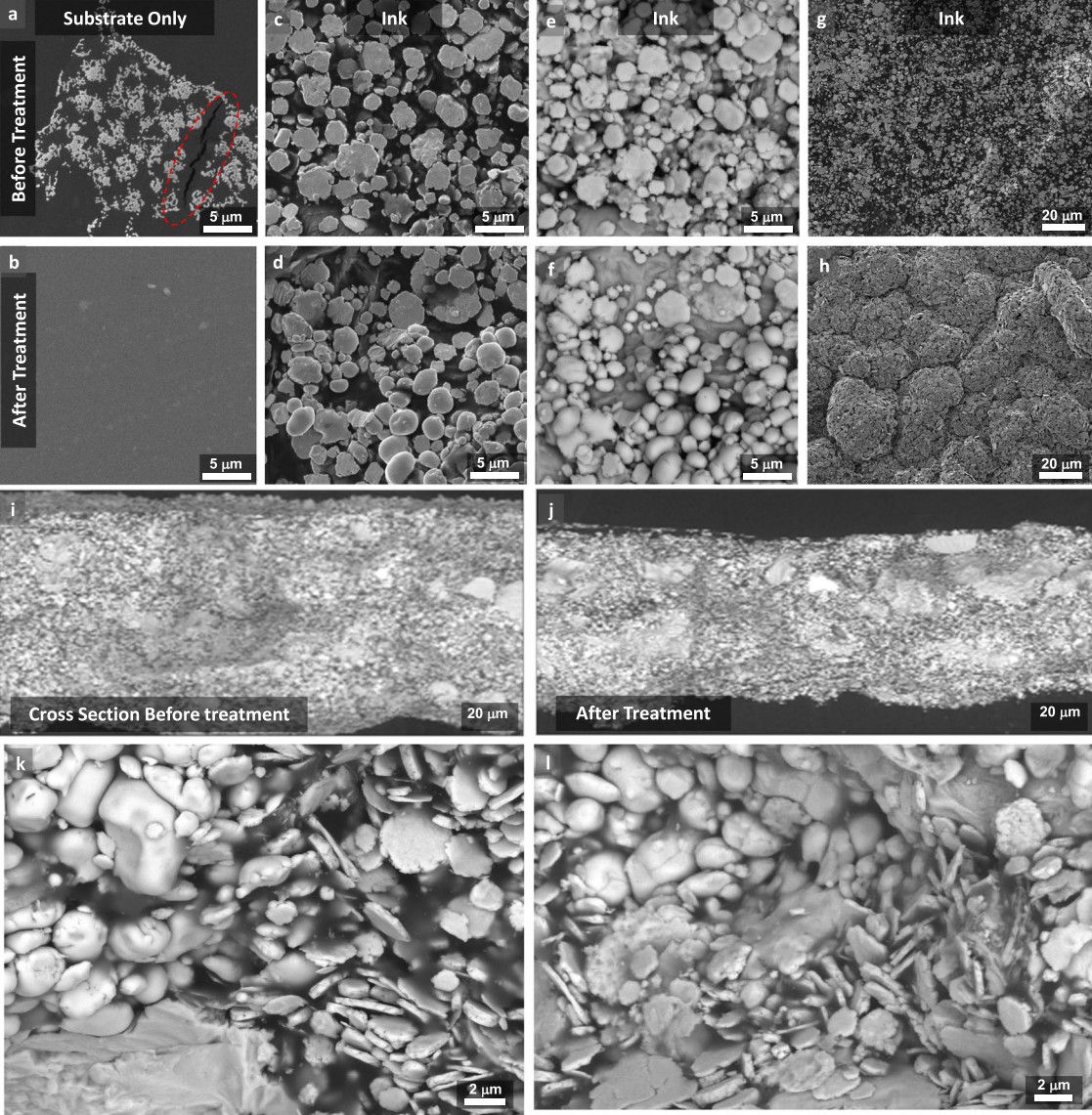

**Fig. 3 Scanning electron microscopy.** SEM of the substrate and the printed ink before and after vapor exposure. SIS substrate before (**a**) and after (**b**) the exposure. The SE images of the ink before (**c**) and after exposure (**d**). The BSE images of the ink before (**e**) and after exposure (**f**). Highly diluted ink spray coated over the textile before (**g**) and after (**h**) vapor exposure. Cross section of SEM images of the printed ink before (**i**, **k**) and after vapor exposure (**j**, **l**).

compared to the SE images. Additional images, as well as EDS analysis, are presented in Supplementary Figs. 11–13, which are in accordance with and demonstrate that more $AgIn_2$ is present on the top surface after the Pol–Gel and the subsequent Gel–Pol transition.

We then spray coated a thin layer of the ink over the textile. Figure 3g, h shows the microstructure of the samples before and after vapor exposure. Here, the morphology of the ink after treatment is very compact, and relatively large raspberry shape clusters (>20 μm) can be seen. More SEM images of this sample can be seen in Supplementary Fig. 8. SEM microscopy from cross section of various samples showed a 10% reduction in the ink thickness (Fig. 3i, j and Supplementary Fig. 5). Higher magnification images (Fig. 3k, l) show that after treatment the Ag flakes, $AgIn_2$, and EGaIn droplets are entangled together in a more compact form when compared to the sample prior to the treatment.

The dark color "blurred" spaces visible in Fig. 3k are SIS polymers. After vapor exposure, the amount of SIS is reduced, and it is also better distributed. During vapor exposure, as the SIS turns

into a gel, microparticles move downward due to the gravity, thus pushing some of the SIS out of the ink. This results in a denser 3D percolating network, thus improving the conductivity. It is however interesting that the ink fully keeps its integrity inside the SIS gel. Videos captured during vapor exposure (Supplementary Video 4) show that the printed traces of the ink can descend or even relocate (in case the surface over which they are placed is not fully level or flat), but the ink always keeps its integrity and does not leave the particles behind.

**Self-soldering, self-healing, and self-coating of circuits**. The reversible Pol–Gel transition facilitates the fabrication of SST-integrated stretchable circuits by eliminating a number of processes. In printed electronics, after the first layer of the circuit is printed, microchip interfacing is performed through a number of steps, including selective deposition of conductive adhesives, placement of the chips, thermal sintering, and encapsulation. The solvent exposure eliminates all these steps and results in a seamless integration of the chips into the circuit in a single step through three mechanisms. During the gel state, the conductive pads of the

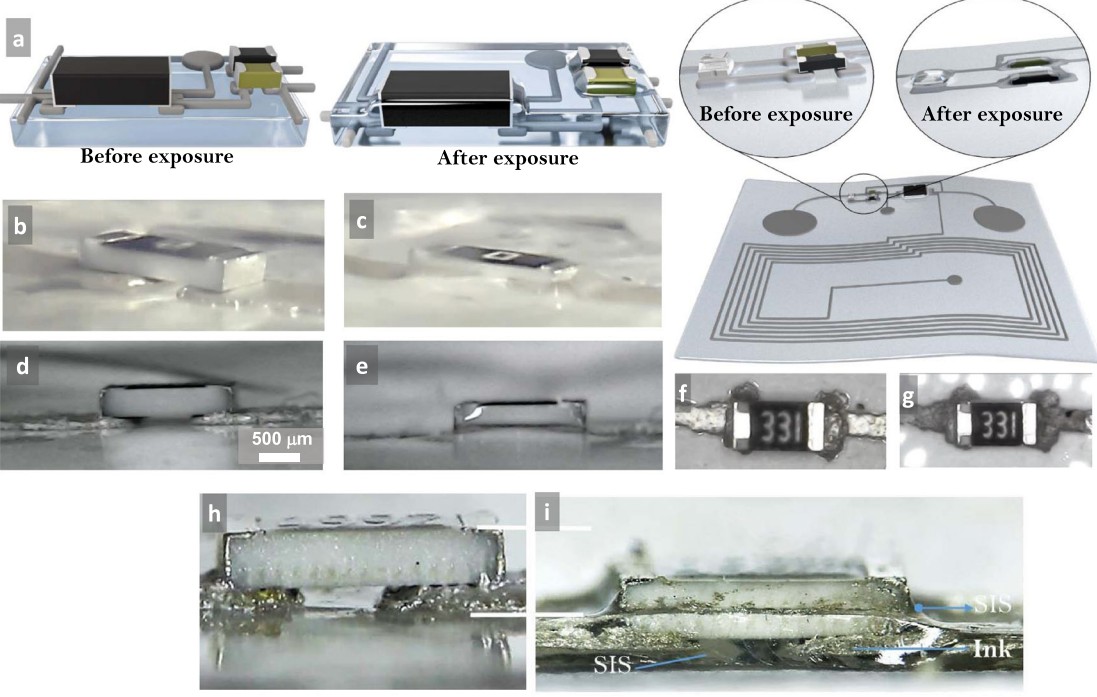

**Fig. 4 Details of the self-soldering process. a** A schematic presentation of the microchip integration into the soft-matter circuit before and after exposure. Snapshot of a resistor component before (**b**) and after vapor exposure (**c**). Note the penetration of the component and the ink into the substrate. Side view of a resistor chip before (**d**) and after (**e**) vapor exposure. The light reflection of the chip after exposure shows the SIS that climbed the walls of the chip due to capillary action. Top view of the resistor chip before (**f**) and after (**g**) vapor exposure. The color difference of the ink demonstrates encapsulation. Magnified side view of the resistor chip before (**h**) and after (**i**) vapor exposure. All of the ink and approximately half of the resistors are inside the substrate. Note also the SIS below the resistor and on the sides.

microchip package penetrate and adhere to the conductive ink, the bottom side of the package adheres to the underlying substrate, and as the chip is penetrating the substrate, it is fully surrounded on all four sides by the adhesive polymer (Supplementary Videos 1–3). Capillary action also seems to contribute to this. Figure 4a shows schematics of the before and after vapor exposure. Figure 4b versus 4c and 4d versus 4e shows samples before and after treatment from two different angles. The light reflection in Fig. 4e is due to the SIS that climbed the wall of the component.

Figure 4h, i shows the side view of a chip before and after exposure, which shows the penetration of the chip into the substrate. It can be seen that a small amount of SIS climbed the sides of the package due to the capillary action. The color change of the printed trace is due to the self-coating of the printed trace by SIS. This can also be seen in Supplementary Video 4.

A thin layer of SIS covers the ink, thus eliminating the need for an additional encapsulation step. This SIS layer not only protects the ink from corrosion and oxidization but is also very resilient to sharp objects (Supplementary Video 6). The thickness of this film depends on the exposure time and vapor intensity. In one test, we also showed that it is possible to coat the microchips. We used a thicker substrate (~2 mm thick) over which we placed an LED and exposed it to the toluene vapor. The circuit was fully coated, including the LED (Supplementary Fig. 10).

Figure 5a demonstrates the schematics of the healing process when subjected to vapor exposure. After a deep cut is made on the printed sample, both the substrate and the ink reconnected efficiently when the Pol–Gel transition is enabled by the vapor exposure. The LM microparticles also contribute to the restoration of electrical conductivity. Figure 5b shows a sample during cutting, after cutting, and after healing. In addition, Fig. 5b shows the sample prior to and after healing under the optical profilometer (Profilm3D 205-0897, Filmetrics).

The healing is very effective in that it not only restores the electrical conductivity but also the circuit can be stretched again after healing (Fig. 5c). Also, Fig. 5c shows the magnified image of a sample subject to >900% strain. As can be seen, the failure happens due to the creation of the holes on the interface between the microchip and the ink, similar to the virgin samples. The healed zone did not contribute to the failure of the sample. The healing process is as well demonstrated in Supplementary Video 7, in which a sample that is deeply cut and healed can successfully withstand the strain. It is also worth mentioning that the reversible Pol–Gel property of BCPs can make these circuits susceptible to damage if exposed to the high density of their solvent vapor for a long time. However, minor contact periods with these solvents are not enough to cause any damage. Besides, low solvent quantities also evaporate very quickly, reducing, even more, the contact time.

In addition to direct digital printing, stretchable circuits can also be produced using laser patterning a full Ag–In–Ga–SIS conductive film or by stencil printing (Fig. 6a). Figure 6b shows an example of an LED mesh display produced by digital printing. Direct digital printing is very convenient and cost-effective. However, due to the mechanical limitations of the extrusion head, when a printing resolution >200 μm is required, laser patterning is preferred.

Figure 6c shows an example of a complex miniature circuit that was produced by laser patterning, using a desktop fiber laser. This circuit integrates a microprocessor, a temperature sensor, and a Bluetooth module, and is able to communicate the body temperature to a smartphone. The toluene vapor method showed to be very effective to attach ICs with different size packaging. Various demo circuits are shown in Supplementary Video 8. Supplementary Fig. 19 shows successful interfacing of a small resistor array chip, where the gap between printed traces was 200 μm, without any short circuit.

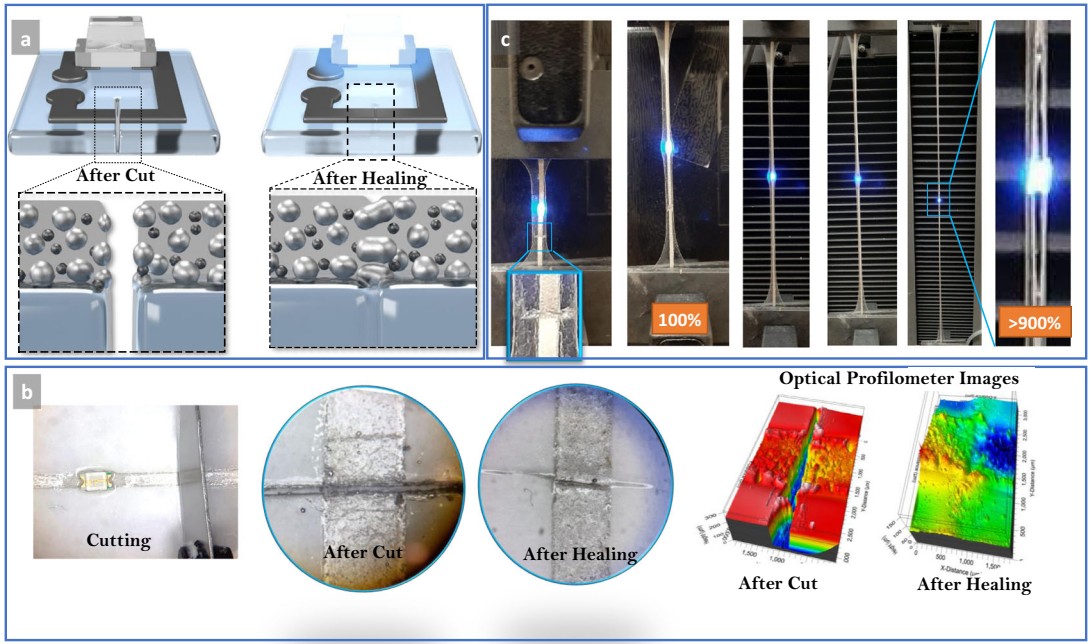

**Fig. 5 The self-healing process. a** Schematics of the healing process. **b** Image from the sample undercut, after a deep cut of the ink and the substrate, and after healing through vapor exposure. Optical profilometer images of the cut before and after healing. **c** A dully cut and healed sample under tensile strain test of 900%, and the magnified image of the sample prior to breaking.

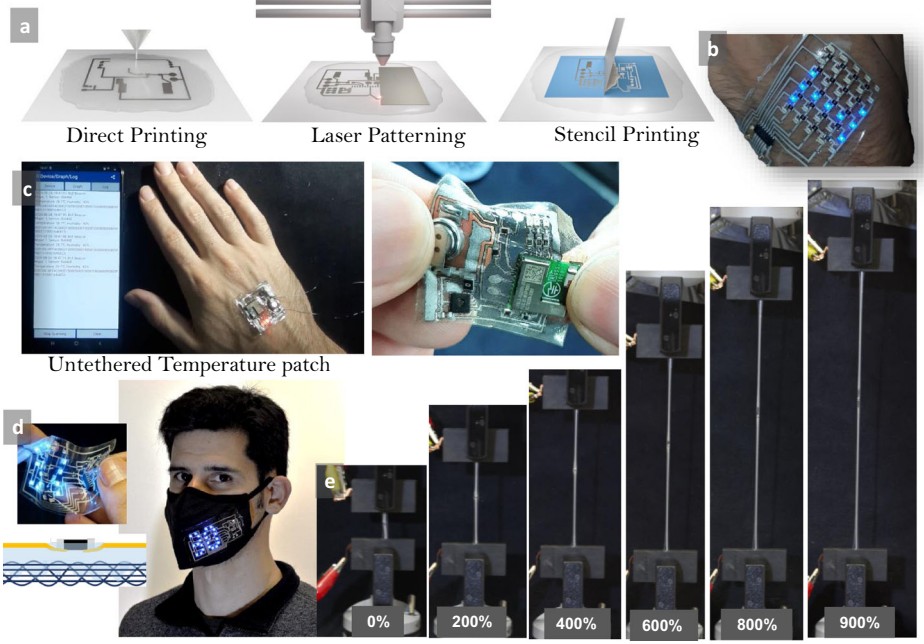

**Fig. 6 Circuit patterning techniques and examples of applications. a** Soft-matter circuit with Ag–In–Ga–SIS ink can be produced by extrusion printing, fiber laser patterning, or stencil printing. **b** Example of an LED display with 60 integrated chips fabricated by extrusion printing and vapor exposure. **c** An untethered wireless temperature patch with integrated Bluetooth chip, temperature and humidity sensor, microprocessor, resistors, and battery, produced through laser patterning and vapor exposure. The patch sends the data to the mobile phone app. **d** A condensed soft-matter circuit with temperature sensor and LED display matrix, transferred to the textile through the same Pol–Gel procedure. During the gel state, the SIS substrate penetrates into the fibers of the e-textile, resulting in seamless integration. **e** Example of a sample with a resistor chip <0–900% strain. The authors affirm that the human research participant (first author P.L.) provided informed consent for publication of the images in (**b–d**).

Finally, another interesting possibility that the Pol–Gel technique provides is the integration of the printed circuit into other porous surfaces such as the textile (Fig. 6d). The circuit is simply placed above the substrate and exposed to the solvent vapor. In the gel state, the substrate penetrates into fibers of the textile and gets fully integrated into the host fibers. Once the external stimulus is removed, the Pol–Gel transition is reversed to Gel–Pol and the circuit is fully integrated into the textile.

This allows extending the application of these printed circuits for scalable fabrication of e-textile, in-mold electronics, and structural electronics.

An example of a temperature monitoring mask is shown in Fig. 6d. An example of a circuit that remained functional at <900% strain is shown in Fig. 6e. Supplementary Video 8 demonstrates various complex circuits with many integrated microchips produced by this technique, including an LED display with 60 packages (LEDs and Resistors), a wireless temperature and humidity monitoring patch with integrated Bluetooth, microcontroller, battery, temperature and humidity sensor, which sends the data wirelessly to a mobile phone application, and an e-skin temperature measurement patch with an LED display

## Discussion

In this study, we exploit the reversible Pol–Gel transition of elastic polymers as a versatile method for the fabrication of microchip-integrated ultra-stretchable circuits. We show a record-breaking maximum strain of >500% prior to electrical failure, a stable behavior for 1000 cycles of 100% strain, and 500 cycles of 400% strain. To achieve the Pol–Gel transition, we proposed a simple vapor exposure technique that enables a phase transition in the SIS substrate and the SIS-containing ink. This technique can not only be used for interfacing the microchips to the printed circuits but it can also improve the electrical and mechanical properties of the circuit. The vapor exposure also results in an interesting self-coating that encapsulates the printed traces with a very resilient SIS-protective layer. If the circuit is cut, vapor exposure heals the circuit effectively to a point that it can survive large strains. Finally, this procedure can be used for the integration of the printed circuit with integrated chips into the textile. Extensive microscopy and elemental analysis were performed in order to analyze the effect of vapor exposure on the substrate, the ink, and the microchip interface. In addition, electromechanical coupling of the samples with and without the microchip was characterized.

To print the circuit, we used an SIS block–copolymer substrate and an SIS–Ag–EGaIn ink. The circuit can be printed using an extrusion printer or patterned using a fiber laser. Unlike previous methods for interfacing microchips that require many fabrication steps, this technique is a simple, single step and is performed at room temperature. The Pol–Gel technique transforms the elastomer substrate and the ink into adhesive gels that conforms to the silicon chips and surrounds the IC body on five sides. Overall, this self-soldering, self-coating, and self-healing technique is a simple and versatile method and an important step toward reducing the complexity of microchip interfacing, and thus for scalable fabrication of microchip-integrated stretchable circuits.

Although SIS is used in this work as the substrate and in the ink formulation, this method can be extended to other polymers that can go through a reversible phase transition through vapor or thermal treatment.

## Methods
**Printing**. Ink cartridges are filled with the AgInGa ink and the printing is performed later using a Voltera V-One. To achieve a uniform circuit, usually two layers of ink are printed. However, we have also successfully tested 14 layers. After printing, the circuits should be left for 2 h at room temperature to ensure solvent evaporation. Alternatively, solvent evaporation can be performed for 30 min at 60 °C.

**Pick and place**. Chips are then placed on the circuit using a pick and place machine (eC-placer Eurocircuits).

**Toluene vapor exposure**. The circuit is then placed in an enclosed chamber that contains a fabric or a paper soaked with toluene, which generates the toluene vapor in the chamber. The treatment time depends on the size of the chamber.

For samples that used electromechanical coupling characterization, the samples are placed in a 0.125 L bottle for 45 min. For the larger circuits, they are placed in a 1 L container for 45 min.

**SIS substrate preparation**. The SIS solution is prepared using a 1:2 ratio of SIS (styrene 14%, Sigma-Aldrich) and toluene. When the mixture becomes a homogeneous solution, it is applied by a thin-film applicator (ZUA 2000, ZEHNTNER) and adjusted at 800 μm over a nonstick silicone paper (RUSPEPA, SP1215-D20-E) from Amazon.

**Electrical and electromechanical characterization**. The dogbones used were fabricated using the die-C ASTM D412 standard. See Supplementary Fig. 1 for the size of the dogbone samples. Dogbones were made by laser patterning (VLS 3.50, Universal Laser Systems Inc.). For electromechanical testing, we used an Instron 5969 with a 100 N load cell and a data acquisition system composed of a multimeter (gwInstek gdm-8351) and 16-bit DAQ (NI USB 6002). Conductivity measurements are performed at the two square leads of each sample using a desktop multimeter (Fluke 45). The conductivity is calculated from the electrical resistance and the conductance is estimated based on the thickness of the printed samples after cross-section analysis with SEM. For electromechanical testing, a 10 Ω SMD resistor (0805, 2.0 mm × 1.2 mm × 0.45 mm) package was interfaced over a 1 mm gap in an ink track through a 45 min vapor treatment.

**Ink synthesis**. The ink is prepared by dissolving SIS in toluene (15 wt% SIS) until a clear solution is obtained. For each 5 g of BCP solution, 6.2 g of Ag flakes (Silflake, Technic) and 15 g of EGaIn are added and mixed using a planetary mixer (Thinky ARE-250) at 2000 r.p.m.

**Laser patterning**. The procedure consists of spreading a conductive ink film over the substrate using a thin-film applicator and then a pulsed fiber laser (1064 nm wavelength) selectively removes ink to isolate the circuit traces. Because the laser wavelength only affects metals, the polymer substrate under the ablation area remains intact. This turns out to be an effective method to fabricate high-resolution circuits with spacing as low as 70 μm, thus enabling the possibility of using smaller IC packages in our applications. However, with this method, large amounts of ink are wasted during the process.

**SEM and elemental analysis**. The surface and cross-section morphologies of the samples were characterized by SEM using an FEI Quanta 400FEG ESEM equipped with an EDAX Genesis X4M, which enabled the elemental analysis by EDS. For the cross-section observation, the samples were immersed for 90 s in liquid nitrogen. This allowed for a clean fracture of the samples to be made through mechanical impact. The micrographs were collected in SE and BSE modes. Full-scan SEM images are provided in the Supplementary information.

**Over the skin patches and publication consent**. Skin patches were tested by the first author P.L. after filling a consent form. The authors affirm that the human research participant (first author P.L.) provided informed consent for publication of the images in Figs. 1i, j and 6b–d.

## Data availability
The authors declare that the main data supporting the findings of this study are available within the article and its Supplementary information. Source data are provided with this paper.

## Code availability
The code used in the example circuits is available upon any reasonable request.

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

## Acknowledgements

This work was partially supported by the Carnegie Mellon-Portugal project WoW (Reference No.: 45913), which had the support of the European Regional Development Fund (ERDF) and Dermotronics (PTDC/EEIROB/31784/2017), financed by the EU Structural & Investment Funds (FEEI) through an operational program of the center region, and the Foundation of Science and Technology (FCT) of Portugal. Funding also came from Add.Additive (POCI-01-0247-FEDER-024533), financed by Regional Development Funds (FEDER), through Programa Operacional Competitividade e Internacionalização (POCI). Access to instruments of TAIL-UC facility supported by the QREN-Mais Centro program ICT_2009_02_012_1890 is gratefully acknowledged.

## Author contributions

P.A.L. and B.S performed experiments; P.A.L. and B.S. and M.T. discussed the results; M.T. and A.T.d.A. supervised research; P.A.L., B.S., and M.T. wrote the manuscript.

## Competing interests

The authors declare no competing interests.
