## [Peer Review File · Nature Communications]

REVIEWER COMMENTS

Reviewer #1 (Remarks to the Author):

This manuscript reports a new method to fabricate stretchable electronics. The results are compelling and suitable for publication in Nature Communications. I have a few minor comments and questions.

Two small typos:

Abstract - simplify simplicity

Bottom of pg 7 - themthe

Fig 1h is very small. Are all the compared techniques on the same substrate?

Could you sync up the colors in Fig 2 to make it easier to follow? For example, in Fig 2d blue is before treatment and yellow is after treatment, but the opposite color assignments are used in Fig 2e.

I don't fully understand how the one-step process of vapor exposure leads to the two-step outcome of both penetrations of the ink and components into the substrate and their self-encapsulation in SIS. Could the authors please clarify how this happens? The authors cite capillary action. Is that what I'm seeing in the Video "self-sealing of the ink"?

The citations in this paper look thorough and complete. I will note a couple of papers that just came out (after this paper was submitted, so there was no way the authors could have cited them or compared their approaches before):

doi.org/10.1038/s41563-021-00921-8

doi.org/10.1038/s41563-020-00902-3

Reviewer #2 (Remarks to the Author):

The manuscript "Reversible Polymer-Gel transition for Ultra-Stretchable Chip-Integrated Circuits: Self-(Soldering, Coating, and Healing)" by Lopes et al presents a one-step process to integrate solid state electronic components on elastic substrates. Essentially a printed circuit is made to sink into a temporary melted elastic substrate. The method is elegant and will be of interest to a broad range of researches in soft/stretchable electronics because it can significantly simplify complex integration steps. Before publication, the manuscript needs improvements. Some points are outlined below:

1. Figure 1. Poorly organised and crowded. A clear diagram of the process concept would be useful here.
2. An explanation is needed of why circuits can survive such high strains. At the SST-elastic interconnect interface a mechanical discontinuity is present which results in high stress concentration. How does the solder cope with this?

3. Figure 2 (a) a log scale would improve clarity.
4. Figure 2. Reporting resistance is of limited use as it is dependent on sample geometry. Reporting relative resistance change would be more informative.
5. The gauge factor of the interconnects should be calculated and reported.
6. Figure 2. Check the order in which panels are discussed/referred to in the text. There is a mistake with (f).
7. Include error when reporting ink conductivity.
8. Figure 5, presentation should be improved. What does i_v and v refer to?
9. Authors should explain about the resilience of the SIS material to common solvents that may accidentally come up in contact with the device when deployed in the field.
10. Additional relevant literature can be included. See examples below:
<https://doi.org/10.1038/s41551-020-00615-7>
<http://dx.doi.org/10.1002/adma.201703817>

Reviewer #3 (Remarks to the Author):

The manuscript by Lopes et al. describes the use of solvent processing to achieve improved adhesion and electronic properties in hybrid printed/pick and place electronics. The solvent annealing process facilitates surface-energy-mediated alignment of the printed traces, good adhesion between the printed electronic components and the rigid electronic components, and healing of fabrication-related defects. The process is simple and could be implemented into many fabrication processes, as demonstrated by the use of DIW printing, screen printing and laser ablation. The manuscript contains relatively little scientific work, but the engineering demonstrations are convincing so that many groups will likely use this process in the future. I recommend that this work should be published in Nature Communications with some minor changes.

Comments

1. The manuscript generally contains many relevant papers, but is missing some of the important literature on this topic, including:
Pick and place electronics: [Valentine, *Adv Mater*, 29:1703817, 2017][Byun et al, *Sci Robot*, 3:eaas9020, 2018][Huang et al, *Nat Electron*, 1:473, 2018]
Solvent-healing of electronics: [Oh et al, *Nature*, 539:411, 2016][Baek et al, *RSC Adv*, 6:98466, 2016]
2. Are there any limitations on the resolution of the traces during the solvent annealing process? For example, if a chip with pins every 200 μm is aligned with the printed traces, does the solvent annealing process ever cause shorts between pins? Or is it self-aligning due to differences in wettability similar to a

typical soldering process?

3. During the solvent annealing process, how does the duration of annealing affect the degree to which the placed circuits sink into the substrate? If the samples are left in the toluene atmosphere for a long time, do the circuits continue to change, or does it reach an equilibrium point when the capillary forces minimize the energy of the system?

Minor comments:

1. P3: "ultra-stretchable interconnects with maximum tensile strength of up to 1000% were demonstrated"

Tensile strength has units of Pa; this is the elongation at break or strain at failure.

2. Figure 1: At least c, e, and g would benefit from having scale bars.

3. Figure 1 h) The references to Cheng et al and Robinson et al don't seem to have a reference in the references section.→→→

4. Figure 2 caption: "a) Max. Strain tolerance"

Extra period.

5. P10 "contributes to improvement of the max. Strain tolerance."

Extra period.

6. P15 "It is important to highlight the role of Block Copolymers (BCP) in the success of this method. As shown in Fig. S18, Styrenic BCPs [...] polymer solvent."

-This comment seems very out of place and would make more sense in the introduction where you are justifying and introducing the approach.

-Block Copolymers does not need to be capitalized

-Styrenic does not need to be capitalized

Reviewer #1

Comments:

This manuscript reports a new method to fabricate stretchable electronics. The results are compelling and suitable for publication in Nature Communications. I have a few minor comments and questions.

Thank you for reviewing our article.

1. Two small typos: Abstract - simplify simplicity. Bottom of pg 7 - them--

>the Thanks for your attention. We have now addressed this problem.

2. Fig 1h is very small. Are all the compared techniques on the same substrate?

The Fig.1h is now clearer. All the compared techniques use polydimethylsiloxane (PDMS) as stretchable substrate. This material still one of the most used polymers in research of stretchable electronics, although more works are recently moving toward the use of block copolymers. But — PDMS doesn't have adhesive properties. Our method takes advantage of using toluene as solvent to reversible transition between polymer state and gel state of SIS polymer.

3. Could you sync up the colors in Fig 2 to make it easier to follow? For example, in Fig 2d blue is before treatment and yellow is after treatment, but the opposite color assignments are used in Fig 2e.

Thanks for this suggestion. The colors from Fig.1d are now corrected.

4. I don't fully understand how the one-step process of vapor exposure leads to the two-step outcome of both penetrations of the ink and components into the substrate and their self-encapsulation in SIS. Could the authors please clarify how this happens? The authors cite capillary action. Is that what I'm seeing in the Video "self-sealing of the ink"?

Capillary action helps, but is not the only reason. Solvent vapors will transform the substrate and ink to a gel like solution. In this case, the components and the ink descend, which let the component to bond to the substrate. At the same time capillary action allows the gel to climb the surfaces of the microchip. We noticed that SIS substrate climbed the microchip walls during the process, the figure below shows a magnified side image of a chip resistor before (h) and after (i) exposure. However, only a thin encapsulation layer is visible after 1 hour exposure, the thickness of the encapsulation depends on the exposure time. Yet, if the exposure time increase a full penetration of ink and components into

the substrate will occur, video S4 – Self-Coating shows an ink trace being exposed for a longer period (~7 hours), the capillary action starts earlier as can be seen by the reflections appearing over the ink due to SIS flow, as the exposure time increase the ink finally submerged totally. In this way, the vapor process will stablish a more robust connection between the chip and the ink and promote encapsulation by the SIS substrate. Figure 1g shows a sample in which not only the ink, but also the microchip are fully encapsulated, but in this case the SIS substrate had to be thicker (3mm thick).

5. The citations in this paper look thorough and complete. I will note a couple of papers that just came out (after this paper was submitted, so there was no way the authors could have cited them or compared their approaches before):

doi.org/10.1038/s41563-021-00921-8

doi.org/10.1038/s41563-020-00902-3

Thank you for bringing these articles to our attention. We have cited both articles in the introduction, as highlighted in the manuscript.

Reviewer #2

Comments:

The manuscript “Reversible Polymer-Gel transition for Ultra-Stretchable Chip-Integrated Circuits: Self-(Soldering, Coating, and Healing)” by Lopes et al presents a one-step process to integrate solid state electronic components on elastic substrates. Essentially a printed circuit is made to sink into a temporary melted elastic substrate. The method is elegant and will be of interest to a broad range of researches in soft/stretchable electronics because it can significantly simplify complex integration steps. Before publication, the manuscript needs improvements. Some points are outlined below:

Thank you for reviewing our article.

1. Figure 1. Poorly organized and crowded. A clear diagram of the process concept would be useful here.

We tried to reorganize Figure 1. Without changing the actual images that constitute figure 1, we tried to readjust to make it clearer, specially Figure 1a that shows the process.

2. An explanation is needed of why circuits can survive such high strains. At the SST-elastic interconnect interface a mechanical discontinuity is present which results in high stress concentration. How does the solder cope with this?

There are several reasons contributing to this. As we had explained in the article:

Unlike the previous works in which the package is only bonded from the bottom surface, this method restricts the package from 5 faces, which contributes to a better mechanical locking, thus a more uniform distribution of the stress at the rigid-soft interface, resulting in a considerably higher strain tolerance compared to previous works. In all samples, mechanical failure happened prior to the electrical failure. We observed that the failure usually starts on the interface between the component and the substrate (**Fig. S3**).

3. Figure 2 (a) a log scale would improve clarity.

The graph is now in log scale.

4. Figure 2. Reporting resistance is of limited use as it is dependent on sample geometry. Reporting relative resistance change would be more informative.

The graph values are now relative R/R_0 .

5. The gauge factor of the interconnects should be calculated and reported.

The gauge factor for the interconnects was calculated from the printed trace samples (**Fig. 2e**) at 200% strain and reported in the manuscript.

6. Figure 2. Check the order in which panels are discussed/referred to in the text. There is a mistake with (f).

The mistake was corrected.

7. Include error when reporting ink conductivity.

The ink conductivity error is now included on the bar graph (**Fig. 2d**)

8. Figure 5, presentation should be improved. What does i_v and v refer to?

Figure 5 legend is now updated.

9. Authors should explain about the resilience of the SIS material to common solvents that may accidentally come up in contact with the device when deployed in the field.

Good Point. We added the following text:

On the other hand, it is also worth mentioning that the reversible POL-GEL property of BCPs, can make these circuit susceptible to damage if exposed to the high density of their solvent vapor for a long time. However, minor contact periods with these solvents are not enough to cause any damage. Besides, low solvent quantities also evaporate very quickly reducing even more the contact time.

10. Additional relevant literature can be included. See examples below:

<https://doi.org/10.1038/s41551-020-00615-7>

<http://dx.doi.org/10.1002/adma.201703817>

Thank you for bringing these articles to our attention. We have cited both articles in the introduction, as highlighted in the manuscript.

Reviewer #3

Comments:

The manuscript by Lopes et al. describes the use of solvent processing to achieve improved adhesion and electronic properties in hybrid printed/pick and place electronics. The solvent annealing process facilitates surface-energy-mediated alignment of the printed traces, good adhesion between the printed electronic components and the rigid electronic components, and healing of fabrication-related defects. The process is simple and could be implemented into many fabrication processes, as demonstrated by the use of DIW printing, screen printing and laser ablation. The manuscript contains relatively little scientific work, but the engineering demonstrations are convincing so that many groups will likely use this process in the future. I recommend that this work should be published in Nature Communications with some minor changes.

Thank you for reviewing our article.

1. The manuscript generally contains many relevant papers, but is missing some of the important literature on this topic, including:

Pick and place electronics: [Valentine, Adv Mater, 29:1703817, 2017][Byun et al, Sci Robot, 3:eaas9020, 2018][Huang et al, Nat Electron, 1:473, 2018]

Solvent-healing of electronics: [Oh et al, Nature, 539:411, 2016][Baek et al, RSC Adv, 6:98466, 2016]

Thank you for bringing these articles to our attention. We have now cited these articles in the introduction, and as well added the following text:

“The healing ability of some polymers networks, is an interesting property that enables repairing damage polymers. To do so, several methods had been explored, including solvent mediated⁶⁰, or solvent vapor exposure⁶¹. Using these techniques, cracks can be healed, and electrical and mechanical properties can be recovered by promoting the entanglement of polymer chains with heat above the glass transition temperature or promoting diffusion of the material using a solvent.”

2. Are there any limitations on the resolution of the traces during the solvent annealing process? For example, if a chip with pins every 200 um is aligned with the printed traces, does the solvent annealing process ever cause shorts between pins? Or is it self-aligning due to differences in wettability similar to a typical soldering process?

Good point. We performed a test using a small resistor array chip, where the gap between printed traces were 200um (which is the limit of most extrusion printers). As can be seen in the below pictures (before and after solvent vapor exposure), the chip penetrates the substrate, and the traces did not move or spread during the process. Profilometer measures were also taken to inspect the spacing between traces. The copolymer from the substrate climbed the chip package and flooded the gap between traces, reducing even more the probability of shorts formation. Although, the circuit must be correctly oriented with respect to ground to avoid circuit movement due to gravitational forces. This image is now added to the SI.

Regarding Self-Aligning: From our experiments, we didn't observe any considerable self-aligning, similar to what happens with common soldering process, or with LM+ HCL process. If the chips are not well aligned at first, the mismatch will remain after the vapor process.

3. During the solvent annealing process, how does the duration of annealing affect the degree to which the placed circuits sink into the substrate? If the samples are left in the toluene atmosphere for a long time, do the circuits continue to change, or does it reach an equilibrium point when the capillary forces minimize the energy of the system?

Increasing the exposure time to solvent vapors makes the circuit to sink further into the substrate. At its extreme, it can sink until it reaches the other end of the substrate. An example of a circuit subjected at longer exposure time can be seen at (Fig. S10), the circuit consisting of a thicker substrate (~2mm thick) was exposed for nearly 7 hours, the conductive traces and chip sunk almost 2 mm deep. The chips weight contributes significantly to the sinking process, since is noticeable that the encapsulation process begins close to chips. The limit of the sinking process does not seem to be depend on an equilibrium point of the capillary forces, is limited by the thickness of the substrate and exposure time.

Minor comments:

1. P3: “ultra-stretchable interconnects with maximum tensile strength of up to 1000% were demonstrated”

Tensile strength has units of Pa; this is the elongation at break or strain at failure.

Corrected

2. Figure 1: At least c, e, and g would benefit from having scale bars.

Corrected

3. Figure 1 h) The references to Cheng et al and Robinson et al don't seem to have a reference in the references section.

Corrected

4. Figure 2 caption: “a) Max. Strain tolerance”

Extra period.

Corrected

5. P10 “contributes to improvement of the max. Strain tolerance.”

Extra period.

Corrected

6. P15 “It is important to highlight the role of Block Copolymers (BCP) in the success of this method. As shown in Fig. S18, Styrenic BCPs [...] polymer solvent.”

-This comment seems very out of place and would make more sense in the introduction where you are justifying and introducing the approach.

We have moved this comment to the introduction, as highlighted in the manuscript.

-Block Copolymers does not need to be capitalized

Corrected

-Styrenic does not need to be capitalized

Corrected

Thanks for your attention. We have now addressed these problems.

We thank the reviewers for their insightful comments and feel that these revisions greatly clarify and strengthen the manuscript.

Sincerely,

Mahmoud Tavakoli

REVIEWERS' COMMENTS

Reviewer #1 (Remarks to the Author):

The authors have mostly addressed my queries. Thank you. I have one remaining question. The authors state that in Fig 1h their comparisons are made to samples on PDMS substrates (presumably Sylgard 184, because it is common). However, their substrate is not Sylgard 184. This sub-figure does not represent a useful comparison. It makes the "present technique" seem hugely advantageous for high-strain applications. Sylgard 184, however, is not a very high strain material and the decision to compare to only this substrate massively biases the figure. Other stretchable electronics have been shown to achieve >1000% strains. I don't think this is central to the claims of the paper. My suggestion to the authors is to either (1) revise the plot to include more relevant comparisons, or (2) delete the plot altogether.

Reviewer #2 (Remarks to the Author):

No further comments
Nice manuscript

Reviewer #3 (Remarks to the Author):

The authors have comprehensively addressed the minor comments of the reviewers, and the manuscript, and the manuscript is ready for submission.

Dear Editor,

Thank you for considering our manuscript for publication in Nature Communications. In response to the comments by the reviewers (in bold) we have made the following revisions:

Reviewer #1 (Remarks to the Author):

Comments:

The authors have mostly addressed my queries. Thank you. I have one remaining question. The authors state that in Fig 1h their comparisons are made to samples on PDMS substrates (presumably Sylgard 184, because it is common). However, their substrate is not Sylgard 184. This sub-figure does not represent a useful comparison. It makes the "present technique" seem hugely advantageous for high-strain applications. Sylgard 184, however, is not a very high strain material and the decision to compare to only this substrate massively biases the figure. Other stretchable electronics have been shown to achieve >1000% strains. I don't think this is central to the claims of the paper. My suggestion to the authors is to either (1) revise the plot to include more relevant comparisons, or (2) delete the plot altogether.

Thank you much for your second round of review. As suggested, we removed figure 1h.

Reviewer #2 (Remarks to the Author):

Comments:

No further comments Nice manuscript

Thank you for reviewing our article.

Reviewer #3 (Remarks to the Author):

Comments:

The authors have comprehensively addressed the minor comments of the reviewers, and the manuscript, and the manuscript is ready for submission.

Thank you for reviewing our article.

Thanks for your attention.

We thank the reviewers for their insightful comments and feel that these revisions greatly clarify and strengthen the manuscript.

Sincerely,

Mahmoud Tavakoli